# Recent Developments in Protein Lactylation in PTSD and CVD: Novel Strategies and Targets

**DOI:** 10.3390/biotech12020038

**Published:** 2023-05-15

**Authors:** Zisis Kozlakidis, Patricia Shi, Ganna Abarbanel, Carolina Klein, Adonis Sfera

**Affiliations:** 1International Agency for Research on Cancer, World Health Organization (IARC/WHO), 69372 Lyon, France; 2Department of Psychiatry, Loma Linda University, Loma Linda, CA 92350, USA; 3Patton State Hospital, University of California, Riverside, CA 92521, USA; 4Napa State Hospital, Napa, CA 94558, USA; 5Department of Psychiatry, University of California, Riverside, CA 92521, USA

**Keywords:** PTSD, CVD, lactylation, hypoxia-inducible factor 1, angiotensin II, interleukin 7

## Abstract

In 1938, Corneille Heymans received the Nobel Prize in physiology for discovering that oxygen sensing in the aortic arch and carotid sinus was mediated by the nervous system. The genetics of this process remained unclear until 1991 when Gregg Semenza while studying erythropoietin, came upon hypoxia-inducible factor 1, for which he obtained the Nobel Prize in 2019. The same year, Yingming Zhao found protein lactylation, a posttranslational modification that can alter the function of hypoxia-inducible factor 1, the master regulator of cellular senescence, a pathology implicated in both post-traumatic stress disorder (PTSD) and cardiovascular disease (CVD). The genetic correlation between PTSD and CVD has been demonstrated by many studies, of which the most recent one utilizes large-scale genetics to estimate the risk factors for these conditions. This study focuses on the role of hypertension and dysfunctional interleukin 7 in PTSD and CVD, the former caused by stress-induced sympathetic arousal and elevated angiotensin II, while the latter links stress to premature endothelial cell senescence and early vascular aging. This review summarizes the recent developments and highlights several novel PTSD and CVD pharmacological targets. They include lactylation of histone and non-histone proteins, along with the related biomolecular actors such as hypoxia-inducible factor 1α, erythropoietin, acid-sensing ion channels, basigin, and Interleukin 7, as well as strategies to delay premature cellular senescence by telomere lengthening and resetting the epigenetic clock.

## 1. Introduction

In response to stressors, cells undergo premature senescence, a state of proliferation arrest, glycolysis-fueled metabolism, and a detrimental secretome, known as senescence-associated secretory phenotype (SASP), which drives organismal aging by disseminating senescence to the neighboring healthy cells [1]. SASP has been implicated in cellular deregulation as it is the precursor of many pathologies, including cancer, while at the same time providing a fertile scientific ground for the discovery of potential new treatments [2]. For example, SASP has been associated with shortened telomere length (STL) and impaired DNA methylation, the epigenetic clock (EpiClock), hallmarks of posttraumatic stress disorder (PTSD), and cardiovascular disease (CVD) [3,4]. These features likely account for the shorter lifespan and early development of age-related diseases demonstrated in many patients with PTSD and CVD [5,6,7]. 

Hypoxia is another important cellular stressor as dysfunctional oxygenation and hypoxia-inducible factor 1 alpha (HIF-1α) lead to premature aging, a characteristic of PTSD and CVD [5,8,9,10]. Indeed, both hypoxia and hyperoxygenation have been associated with neuropsychiatric pathology, and several studies have shown that hyperbaric oxygen or treatment with erythropoietin (EPO) can be efficacious in several psychiatric disorders, including PTSD and traumatic brain disorder (TBI) [11,12,13,14,15,16] (NCT00525863).

At the subcellular level, hypoxia induces endoplasmic reticulum (ER) stress, a common PTSD and CVD pathology, suggesting that psychological and biological stressors can contribute to molecular stress [17,18,19,20,21]. In this regard, it is tempting to speculate that ER stress and the unfolded protein response (UPR) may mirror the organismal stress. Moreover, hypoxia upregulates oxytocin (OXT), while dysfunctional HIF-1α lowers the levels of this protective hormone, predisposing to PTSD and CVD [22,23].

A recent study has identified hypertension as the common genetic risk factor for PTSD and CVD, indicating that sympathetic arousal and elevated angiotensin II (ANG II) levels contribute to these disorders [3,24]. Moreover, the connection between high blood pressure and hypoxia has been well documented in pulmonary arterial hypertension (PAH), a condition marked by dysfunctional HIF-1α and premature endothelial aging [25,26,27]. Interestingly, about half of PAH patients also experience depression, anxiety, or panic disorder, indicating that this disease may be utilized as a working model of PTSD/CVD [28,29].

Under normal circumstances, endothelial cells (ECs) utilize aerobic glycolysis or the Warburg effect for their bioenergetic need, deriving over 80% of adenosine triphosphate (ATP) from lactate despite oxygen availability [30,31,32]. As senescent ECs upregulate lactate even more, excessive glycolysis likely promotes protein lactylation, predisposing to pathology, including PTSD and CVD [33,34,35]. For example, lactylation of HIF-1 α may disable this protein, causing premature EC senescence, which in turn increases vascular permeability, contributing to the pathogenesis of several cardiovascular and neuropsychiatric illnesses [36,37,38].

Dysfunctional interleukin 7 (IL-7), a cytokine known for shifting cellular metabolism from oxidative phosphorylation (OXPHOS) to aerobic glycolysis or Warburg effect, has been associated with PAH, indicating that upregulated lactate serves as a lactylation substrate [39,40]. Lactylation is a posttranslational modification (PTM) of histone and non-histone proteins which alters the physiological function of many biomolecules, including HIF-1α [38].

The association of IL-7 with lactylation is significant as, under normal circumstances, this cytokine opposes cellular senescence and lowers blood pressure [41], suggesting that its dysfunction may drive PTSD and CVD, validating the study of Seligovski et al. [24,39,42,43,44]. In this regard, impaired IL-7 signaling with IL-7 receptors (IL-7R) on ECs, likely contributes to PTSD and CVD by promoting early vascular aging (EVA) and premature ECs senescence [5,34,43,44,45,46,47,48,49]. In contrast, upregulated HIF-1α exerts cardioprotective and neuroprotective effects and can alleviate many PTSD symptoms [50,51,52]. For example, preclinical studies have shown that intermittent hypoxia may ameliorate both PTSD and CVD, further connecting these conditions to oxygen disturbances [53,54].

Several studies have reported that HIF-1α upregulates erythropoietin (EPO), a hormone with demonstrated antidepressant and anxiolytic properties, highlighting the cardio and neuroprotective effect of this protein [55,56,57,58]. Indeed, as EPO counteracts hypoxia by increasing the systemic oxygen-carrying capacity, it protects neuronal cells and cardiomyocytes from ischemia–reperfusion injury [50,51,52,53,54,55,56,57,58,59,60]. Conversely, impaired EPO signaling, documented in PTSD and CVD, points to this pathway as a druggable target for both conditions [55,61]. Under normal circumstances, IL-7 upregulates EPO, reducing the adverse effects of psychological stress, and further emphasizing the beneficial role of these molecules in PTSD and CVD [55,62]. Moreover, depleted IL-7 has been associated with the atrophy of the thymus gland, a characteristic of PTSD and CVD, suggesting that supplementation with IL-7 could ameliorate these pathologies [63,64,65,66]. Furthermore, photobiomodulation (PBM), a non-thermal red-light therapy, has been shown to reverse thymic involution, indicating potential benefits in PTSD and CVD [67].

Although poorly defined, the existence of an intra-thymic OXT system has been known for at least four decades, linking this gland to OXT depletion [68,69,70]. Indeed, intranasal OXT administration was found beneficial for neuropsychiatric pathologies, including PTSD and psychotic polydipsia, linking these conditions to IL-7 [71,72,73,74,75]. As IL-7 has also been implicated in neuromyelitis optica (NMO), a condition marked by antibodies against aquaporin 4 (AQP-4) water channels, IL-7 may protect not only from PTSD and CVD but also from NMO, and the related condition, multiple sclerosis (MS) [76,77,78,79].

This review summarizes the recent developments and highlights novel pharmacological targets for PTSD and CVD, including lactylation of histone and non-histone proteins, along with the related biomolecular actors, HIF-1α, EPO, ASC1 channels, CD147, and IL-7 (Figure 1). Potential interventions and new strategies are also discussed.

## 2. Human Stress Response: A Quick Reminder

Biological and psychological stressors of sufficient intensity may activate organismal responses which aim at restoring homeostasis by inducing physiological or behavioral changes [80]. For example, stressors, such as medical illnesses or social rejection, activate the sympathetic-adreno-medullar (SAM) system, inducing immediate adaptation as well as delayed hypothalamus-pituitary-adrenal (HPA) responses [81,82]. In the central nervous system (CNS), biological and psychological stressors are processed by different pathways, which are believed to converge at the level of the amygdala [83,84].

Amygdala, a sensor for both biological and psychological stressors, functions to activate the HPA axis and restore homeostasis [85]. Amygdala plays a major role in CVD, and the activity of this nucleus is directly correlated with the development of heart disease [86]. The basolateral nucleus of the amygdala (BLA) processes primarily psychological stressors, including aversive stimuli, while the central nucleus of the amygdala (CeA) responds to biological input, such as blood pressure or redox imbalance [87,88,89,90]. However, hypertension due to ANG II-mediated fear-behavior is also processed by the CeA, blurring the boundaries between the stressor types [91,92].

OXT is an established inhibitor of stress-related amygdalar activation, thus a negative regulator of PTSD and CVD, while ANG II has the opposite effect [86,93,94]. Human ECs express abundant OXT and AVP receptors which were implicated in vasodilation, thus benefiting patients with hypertension, a major risk factor for both PTSD and CVD [95].

OXT is produced primarily in the magnocellular neurosecretory cells of the paraventricular and supraoptic nuclei of the hypothalamus and is transported to the posterior pituitary, from where it migrates into the systemic circulation [96]. Smaller sources of OXT, such as the thymus and bone marrow, modulate immune responses, probably mediating the psychosomatic symptoms, such as the sickness behavior [69]. The magnocellular neurosecretory cells project to the amygdala, striatum, hypothalamus, hippocampus, and nucleus accumbens, areas previously implicated in PTSD and other neuropsychiatric illnesses [97,98,99,100,101]. The lateral nucleus of CeA expresses OXT receptors (OXTRs) which are activated by the inwardly rectifying K+ channels (Kir4.1), proteins involved in amygdalar inhibition [102]. Kir4.1 are co-localized with AQP4 water channels which play a major role in cardiovascular homeostasis and synaptic plasticity, probably contributing to PTSD-associated traumatic hypermnesia [103,104,105,106].

## 3. Protein Lactylation, the Warburg Effect 2.0

In the 1920s, Otto Warburg studied the metabolism of cancer cells and discovered that even in the presence of oxygen, malignancies satisfy their energetic requirement by fermenting glucose to lactate instead of carbon dioxide [107]. Lactate was believed to be a metabolic waste product; however, several studies have shown that it can function as a signaling molecule, participating in numerous physiological processes, ranging from immunity to inflammation and gene expression [108]. Lactate-induced epigenetic changes can promote or inhibit protein expression, potentially triggering pathology [109]. For example, the high mobility group box 1 (HMGB1) and HIF1α are lactylation substrates that have previously been associated with tumorigenesis, linking lactylation to malignant transformation [38,109,110]. The metabolic shift from OXPHOS to glycolysis, observed in both PTSD and CVD, may account for lactate-induced anxiety, a phenomenon known since the 1980s but poorly defined previously [111,112,113,114]. Indeed, the molecular underpinnings of PTSD activation by lactate infusion remained unclear until the discovery of acid-sensing ion channels (ASICs) and protein lactylation [109,112,115]. Upregulated lactate and HIF1α lactylation have been shown to promote premature EC senescence, a pathological driver of both CVD and PTSD [116,117]. Under physiological circumstances, HIF1α is a negative regulator of both EC senescence and lactylation; however, loss of this molecule may contribute to EVA and CVD [27,118]. Furthermore, preclinical studies have implicated lactylation in neuropsychiatric pathology, including anxiety, stress, fear, avoidant behavior, and social defeat, emphasizing a probable link to PTSD [35,116,119,120]. These findings are summarized in Figure 2.

### 3.1. Virus-Induced PTSD and CVD

Several pathogens, including SARS-CoV-2, have been associated with a high prevalence of PTSD and CVD, suggesting that a better understanding of virus-induced PTSD (viPTSD) may shed light on the pathogenesis of both conditions [121,122]. The COVID-19 pandemic has highlighted a novel phenomenon, viPTSD, an entity that can be more prevalent than the trauma-related syndrome encountered in special populations, such as combat veterans. For example, new epidemiological data show a PTSD prevalence of 30.2% in COVID-19 survivors, 39.5% in Ebola, and 34% in HIV, numbers that exceed the 8% found in the general population and 16% in war veterans [123,124,125].

We surmise that excessive lactylation may account for the high prevalence of viPTSD as pathogens often establish hospitable microenvironments in host cells by usurping HIF-1α and the Warburg effect [10,27,126]. Elevated lactate levels can also activate ASIC1 channels, triggering PTSD symptoms by a different mechanism [127]. In contrast, IL-7/IL-7R signaling opposes senescence by upregulating HIF-1α and activating the thymus [128].

### 3.2. Virus-Induced Senescence (VIS)

Several viruses, including SARS-CoV-2, have been known to promote premature cellular aging, known as virus-induced senescence (VIS), a phenotype indistinguishable from other forms of senescence [129,130]. Cellular senescence is an antitumor program marked by replication arrest, an active glycolysis-mediated metabolism, and a toxic SASP secretome.

In our previous work, we hypothesized that COVID-19 triggers EC senescence and EVA by blocking angiotensin-converting enzyme-2 (ACE-2), leading to ANG II accumulation [131]. This hypothesis was validated in vitro a few months later [132] and, at present, we hypothesize further that ANG II upregulation is also caused by deficient endoplasmic reticulum aminopeptidase 1 and 2 (ERAP1 and ERAP2), which break down ANG II into ANG III and ANG IV (Figure 3). In addition, others have shown that ANG II disrupts IL-7Rs, enabling viral thriving by inducing immune senescence [133].

ECs are major producers of brain-derived neurotrophic factor (BDNF), a nerotrophin depleted in PTSD, suggesting that senescent endothelia may predispose to this syndrome by depleting BDNF [134]. Furthermore, senescent ECs disrupt the blood–brain barrier (BBB), allowing molecules from the systemic circulation, such as ANG II and lactate, to activate ASIC1, inducing fear-related behaviors [131,135]. ASICs are lactate-activated channels that match the CNS work with energy requirements; however, excessive glycolysis and ASIC activation may trigger anxiety and PTSD [52,115,136,137].

Taken together, viPTSD operates by promoting premature vascular aging, which increases endothelial permeability, allowing circulatory molecules and toxins to access the brain, activating the stress-processing areas, and triggering PTSD.

## 4. From Cellular to Organismal Stress

At the subcellular level, psychological and biological stressors likely merge on the endoplasmic reticulum (ER), inducing molecular stress by activating the unfolded protein response (UPR) [138,139]. For example, ER stress has been associated with major depressive disorder (MDD), PTSD, neurodegeneration, and CVD, bridging not only the gap between these pathologies but also the molecular and organismal stress [18,140,141]. For example, dysfunctional ANG II signaling with angiotensin 2 type 1 receptors (AT1Rs), documented in hypertension, induces ER stress, leading to premature EC senescence and EVA [140,142,143,144,145].

Under normal circumstances, unfolded or misfolded proteins are transported from the ER into the cytosol to be degraded by the proteasome. Accumulation of misfolded proteins and UPR activation in response to biological or psychological stressors may link molecular and organismal aging, while on the other hand, identifying these pathways as pharmacological targets [146]. Indeed, targeting the UPR comprises a novel strategy in the treatment of cardiovascular and neuropsychiatric illnesses, including PTSD [147,148,149,150,151] (see the section on potential interventions). Interestingly, ER stress and UPR activation upregulate IL-7, further connecting cellular stress with PTSD and CVD [152]. Table 1 provides a summary view of those observations.

## 5. The Mystery of ERAP2

The ER harbors ERAP1 and ERAP2, zinc-metalloproteases, which have been implicated in CVD and PTSD (via OXT depletion) [133,163,164]. Indeed, ERAP2 functions as an oxytocinase, breaking down OXN and terminating its action, suggesting that a dysfunction of this enzyme may predispose it to PTSD [133]. OXT is a prosocial hormone with cardio and neuroprotective properties, which under pathological circumstances, was associated with fibromyalgia, PTSD, and CVD [97,165,166,167]. Several studies have reported that ERAP1 and ERAP2 hydrolyze ANG II into ANG III and ANG IV, contributing to the rapid clearance of this toxic peptide (Figure 3). Interestingly, ERAP2 maintains two haplotypes, a long and a short one, of which the latter can be decayed, accounting for approximately 25% of the population which does not express this protein [133].

ERAP1 and ERAP2 single nucleotide polymorphisms (SNPs) have been connected to autoimmune disorders as well as to the susceptibility to bacterial and viral infections, including COVID-19 [168,169]. This may be significant since patients with autoimmune disorders are more susceptible to both PTSD and CVD [170,171]. Indeed, thymic ERAP2 has been associated with autoimmune pathology and susceptibility to PTSD and CVD [169]. Furthermore, ERAP2, an ANG IV agonist, has been shown to upregulate the amygdalar OXT, emphasizing another protective mechanism [172]. ERAPs also protect against hypertension and cellular senescence by promptly hydrolyzing ANG II, terminating its action [173] (summarized in Figure 3). As certain ERAP polymorphisms may lead to chronic, less severe infection, this mechanism may explain the beneficial effect of these proteins in SARS-CoV-2, influenza, and hepatitis C [174,175,176,177]. Interestingly, a new study has reported that ERAP2 likely decreased the vulnerability to Yersinia pestis during the Middle Ages epidemic in Europe, the Middle East, and Northern Africa, emphasizing the antimicrobial properties of this protein [178]. The same authors surmise that ERAP2 resilience to plague may have contributed to the development of modern diseases, including autoimmunity [133,179,180]. We surmise that ERAP2 protects against PTSD and CVD and that loss of this protein drives thymus-mediated autoimmunity, predisposing to these pathologies [181,182].

ERAP1 and ERAP2 proteins reside in the ER and are, therefore, susceptible to ER stress, further linking these enzymes to PTSD and CVD [18,183,184,185]. Indeed, ERAP2 activates autophagy and UPR, suggesting that loss of this protein may also predispose to PTSD [158,186]. For this reason, we believe that:A recombinant ERAP2 may be beneficial for the treatment of PTSD and CVD;ERAP2 may be a viable marker for PTSD vulnerability.

Together, these data highlight several mechanisms linking ERAP2 loss to PTSD and CVD, along with autoimmunity, ER stress, impaired autophagy, thymic involution, dysfunctional OXT processing, and loss of ANG IV.

## 6. Potential Interventions

In this section, we take a closer look at several novel and potentially druggable targets in PTSD and CVD.

### 6.1. OXT

In the CNS, OXT upregulates gamma-aminobutyric acid (GABA), dampening the amygdalar response to stress [187,188]. In addition, OXT modulates the mesocorticolimbic release of dopamine, a neurotransmitter implicated in reward and motivation [189]. Human and animal studies have shown that excessive social isolation lowers OXT levels, removing a major PTSD/CVD protective factor [190,191,192]. Furthermore, as OXT protects against COVID-19 and perhaps long COVID, socialization restrictions during the pandemic may have been counterproductive [193,194,195]. Interestingly, gut microbiota, including Lactobacillus reuteri upregulates OXT, suggesting that supplementation with this probiotic may be beneficial for patients with PTSD/CVD [196,197].

### 6.2. HIF-1α

Tissue hypoxia upregulates HIF-1α, a protective transcription factor that exerts antidepressant effects, lowers glycolysis, and upregulates OXPHOS [198]. Intermittent hypoxia and hyperbaric oxygen have shown promising results in PTSD and CVD, highlighting the role of dysfunctional oxygenation in these disorders [199,200,201].

Adenosine, acting via the A2B receptor (A2BR), upregulates HIF-1α even in the absence of hypoxia, suggesting a druggable PTSD and CVD target [202]. Indeed, PSB-603, an A2BR antagonist, has been found to reduce both obesity and the effects of aging, indicating potential beneficial effects for ECs senescence, EVA, and thymic involution, associated with PTSD and CVD [203,204]. In addition, A2BRs modulate EPO synthesis, a hormone endowed with antidepressant and anxiolytic properties of its own [205,206].

### 6.3. ASIC1 Blockers

ASIC channels, discovered in 1997, are expressed in the central and peripheral nervous systems, where they exert neuroprotective effects [207]. However, excessive ASIC1 opening in response to lactate may trigger pathology, including cancer, chronic pain, anxiety, and PTSD [208,209,210]. In humans, the ASIC1 ortholog, amiloride-sensitive cation channel 2 (ACCN2), is expressed abundantly in the amygdala and the dopaminergic neurons of substantia nigra, areas implicated in neuropathology [211,212]. Since ACCN2 risk alleles, rs685012 and rs10875995 have been associated with panic disorder, manipulation of these proteins may be beneficial for PTSD patients [213].

The inability to cross the BBB is a major disadvantage of ACCN2 blockers; however, a few compounds do enter the CNS. They are as follows:-Amiloride, a potassium-sparing diuretic, which exerts efficacy at the pH of 6.5 and lowers blood pressure, is an established PTSD/CVD risk factor [214]. In our previous work, we as well as other groups, have suggested that amiloride be tested for PTSD [215,216].-C5B, a synthetic ASIC1 inhibitor, crosses the BBB and protects against neuronal apoptosis, indicating potential benefits for PTSD and CVD [216].-Sevanol, a natural lignan extracted from Thymus armeniacus, inhibits ASIC3 and ASIC1, exhibiting potent analgesic and anti-inflammatory properties, indicating potential usefulness in PTSD and CVD [217,218].-The natural flavonoid, Epigallocatechin gallate is a potent inhibitor of ASIC3, implicated in anxiety, pain, and insulin resistance, suggesting potential PTSD effectiveness [219,220].

### 6.4. Lactylation

Lactate plays a major role in the acidification of the tumor microenvironment, while in PTSD, it opens ACCN2, likely exacerbating PTSD symptoms. Lactylation, a PTM which can epigenetically activate or silence gene expression, may play a key role in the pathogenesis of both PTSD and CVD via HIF1α lactylation and functional loss [221]. Aside from its role in PTSD and CVD, HIF1α lactylation was shown to facilitate angiogenesis, exacerbating the invasiveness of some cancers [38]. Moreover, lactylation of membrane-organizing extension spike protein (MOESIN), a protector of cardiomyocytes and neurons, has been associated with premature EC senescence, emphasizing a new PTSD/CVD target [222,223].

Lactylation inhibitors are compounds currently in clinical trials for cancer. However, a small number of these agents exert properties potentially beneficial for PTSD or CVD. Among these, the most significant are lactate dehydrogenase (LDH) inhibitors and demethylzeylasteral (DML).

LDH inhibitors: LDH catalyzes the reversible conversion of lactate to pyruvate and the reduction of NAD+ to NADH. Therefore, inhibition of this enzyme lowers the Warburg effect, likely compelling cells to rely on OXPHOS. Elevated LDH is a risk factor for both PTSD and CVD (measured by Framingham risk score) [224,225]. Indeed, the role of LDH in psychological stress, known for the past four decades, was confirmed by numerous studies; however, the inhibitors of this enzyme and adverse effects have been poorly characterized [226,227,228]. At present, several LDH inhibitors affecting the epigenome may be relevant for PTSD and CVD, including:-5-aminolevulinic acid (5-ALA), which exerts beneficial effects in CVD along with the amelioration of mood, fatigue, and sleep, suggesting potential usefulness in PTSD [229,230,231].-Oxamate has been found helpful for both CVD and psychological stress, indicating potential PTSD benefits [232].-Quinoline 3-sulfonamides, is a novel LDH inhibitor that reverses aerobic glycolysis in cancer cells, and various quinoline derivatives have been tested as anticancer agents [233,234]. Quinolines have been known for their antidepressant and antipsychotic properties, as aripiprazole is derived from these compounds [235]. Several studies have demonstrated the beneficial effect of aripiprazole on PTSD, suggesting that these drugs should be further interrogated [236,237].-Galloflavin is a newer LDH inhibitor that, to our knowledge, has not been assessed for CVD or PTSD, but its pharmacological profile suggests potential efficacy.

Demethylzeylasteral (DML): a compound isolated from the plant *Tripterygium wilfordii*, is a traditional Chinese herbal medicine that inhibits histone lactylation, suggesting beneficial PTSD effects [238]. In this regard, triptolide, an extract of this plant, was shown to lower chronic pain and slow atherosclerosis progression. In addition, triptolide possesses antidepressant properties, suggesting a beneficial effect on PTSD [239].

Glycolysis inhibitors: including WZB117, STF31, or BAY 876, block glucose transporter 1 (GLUT-1), suppressing the Warburg effect in cancer cells [240]. However, blocking GLUT-1 has been associated with severe adverse effects, including seizures, as seen in GlLUT-1 deficiency syndrome (GLUT-1-DS), rendering these agents unsuitable for PTSD or CVD [241]. A summary of the potential target for future interventions in PTSD and CVD is shown in Table 2.

### 6.5. Interleukin 7

IL-7 prolongs the lifespan of T-cells by inducing the activity of telomerase to prevent loss of telomere length. As premature senescence and EVA drive PTSD and CVD, IL-7 supplementation could be beneficial for these conditions [249]. Along this line, GX-17, a homodimeric IL-7, currently in Phase 1b/2 for lymphopenia and cancer, appears to possess properties suitable for PTSD and CVD (NCT02860715) (NCT03752723) [247,248].

A negative regulator of IL-7, aryl hydrocarbon receptor (Ahr), has recently been associated with PTSD and CVD, indicating a druggable target [250,251]. Moreover, the Ahr repressor (AHRR) gene was linked to both PTSD and CVD, suggesting that epigenetic interventions at this level could benefit both conditions [252,253].

### 6.6. Autophagy and UPR

Psychological and biological stressors seem to merge at the subcellular level, disrupting autophagy and UPR. Therefore, targeting autophagy could be a viable strategy for treating PTSD and CVD [254,255]. Indeed, many currently utilized antidepressant drugs enhance autophagy, and the rapid antidepressant action of ketamine has been attributed to this effect [242,243,244,245]. There are also other autophagy activators to be considered. For example, gastrodin induces lysosomal biogenesis and autophagy, suggesting beneficial effects for both PTSD and CVD [256,257,258]. Moreover, as ERAP2 functions as an oxytocinase, recombinant ERAP2 could be beneficial for patients with CVD and PTSD [246].

There are numerous natural and synthetic autophagy activators; however, here, we focus primarily on those preventing or reversing thymic involution and/or lactylation.

AC-73 is a small molecule, an inhibitor of basigin (BSG) or cluster of differentiation 147 (CD147). Basigin, a matrix metalloproteinase inducer, is expressed on the cell surface and mitochondrial inner membrane. During the COVID-19 pandemic, this protein drew the attention of researchers and clinicians as it was identified as an alternative entry portal for the SARS-CoV-2 virus as well as *Plasmodium malariae* [259,260]. In the mitochondrial inner membrane, CD147 is colocalized with LDH, and lactate monocarboxylate transporter 1 (MCT1), linking this protein to the Warburg effect [261]. It has been shown that psychological stress upregulates CD147, which in turn may contribute to cardiac hypertrophy [262,263]. In addition, tissue hypoxia-upregulated CD147 promotes thymic involution, triggering cellular and immune senescence [264,265] (Figure 4).

There is a significant knowledge gap regarding the cellular effects of potential PTSD/CVD agents described above; therefore, further work is required to evaluate the pharmacodynamics, pharmacokinetics, and safety levels of these compounds.

### 6.7. Significance for the Field and Novel Targets

Although identified as a pathology during WWI when PTSD was known as “shell shock”, today, there is a paucity of treatments for this condition [266]. The PTSD cardiovascular involvement was hinted at by the term “soldier’s heart”, used even earlier, during the American Civil War, to describe feelings of anxiety, tension, and being “on edge”.

We have discussed in other articles premature endothelial senescence in PTSD and advocated for different approaches than those utilized today that lack efficacy in most patients [267]. For this reason, we focus primarily on ECs as they derive 80% of their energy from aerobic glycolysis and rarely utilize oxidative phosphorylation (OXFOS) [267]. In contrast, other brain cells, such as resting microglia, rely on OXFOS, while activated microglia shift their metabolism to aerobic glycolysis [268]. Astrocytes are primarily glycolytic and generate lactate that is shuttled to the neurons for their energy needs [269]. Neuroimaging studies have demonstrated that activated areas of the brain upregulate aerobic glycolysis while aging brains decrease this metabolic modality [270]. In PTSD, due to premature cellular aging, there is less activation or even deactivation in the ventromedial prefrontal cortex during trauma-induced script-driven imagery [271].

Although treated primarily by psychiatrists, PTSD is a systemic disease that has been associated with numerous medical problems, including chronic pain, hypertension, fatigue, muscle tension, headaches as well as metabolic syndrome, and immune dysfunction [272,273,274]. As systemic disorders call for systemic interventions, epigenetic therapy to lower the lactylation of histone proteins seems to be an adequate approach.

The treatment targets, including lowering cellular senescence, reversing thymic involution, and enhancing EPO and IL-7, may lower lactylation without totally inhibiting lactate, a metabolic modality utilized by healthy brains. For example, senolytic drugs, including dasatinib, quercetin, and rapamycin, oppose cellular senescence, a phenotype known to upregulate lactylation, especially in ECs [275,276]. Other systemic approaches include ASIC1a and 3-phosphoinositide-dependent protein kinase 1 (PDK1) inhibitors, agents that can reverse cellular senescence and showed beneficial effects in CVD and PTSD [277].

## 7. Conclusions

Life on Earth depends on oxygen, and the brain and heart are the highest consumers of this gas. For this reason, the cardiovascular system and the neuraxis are extremely sensitive to dysfunctional oxygenation. Due to the high energy demand, hypoxia, a danger signal, activates glycolysis, an oxygen-sparing metabolic modality, likely altering cerebral and cardiac function. Lower oxygen concentration triggers anxiety, while chronic hypoxia could lead to persistent feelings of apprehension, which in patients with PTSD can activate the core symptoms.

In compensating for hypoxia, the HIF-1α-EPO axis upregulates the systemic oxygen-carrying capacity, attempting to restore homeostasis. To preserve oxygen, OXT is lowered (as it upregulates the respiratory rate), and cellular housekeeping functions, including autophagy and UPR, are placed on hold. Furthermore, to conserve energy, cellular replication is stopped by the activation of cellular senescence, contributing to premature organismal aging.

Lactylation hinders the HIF-1α compensatory efforts by utilizing this molecule as a substrate, leading to functional loss. Psychological stress, activation of CD147, cardiac hypertrophy, thymic involution, and premature endothelial senescence increase BBB permeability allowing peripheral molecules, including lactate, to access the brain and induce PTSD symptoms. In contrast, the IL-7-thymus axis, a protective system, opposes senescence by several mechanisms, including telomere elongation, the EpiClock adjustment, and thymus reactivation.

Various approaches have been utilized to improve cellular respiration and the associated symptoms, and here we have emphasized several potential strategies which we believe are worth further investigation.

## Figures and Tables

**Figure 1 biotech-12-00038-f001:**
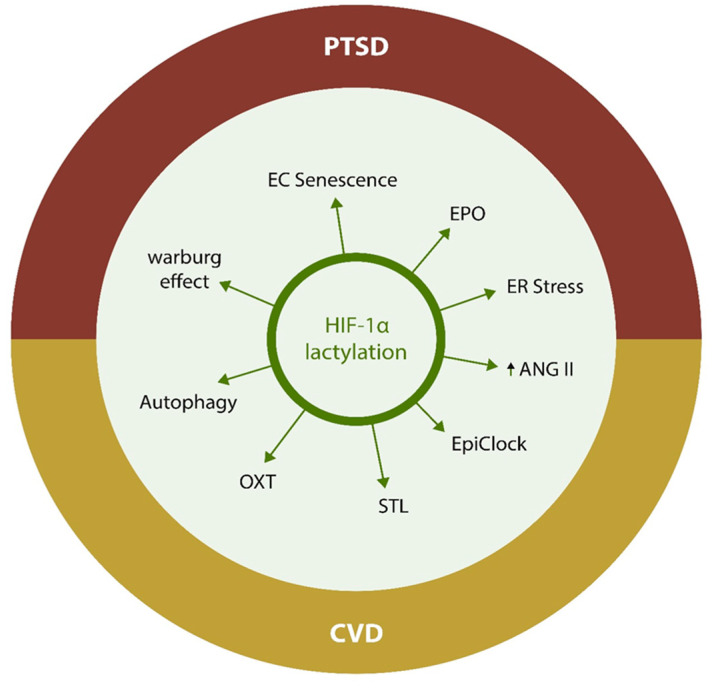
Common pathophysiological mechanisms of PTSD and CVD. Upregulated glycolysis in senescent endothelial cells generate excessive lactate, which serves as a lactylation substrate, triggering a downstream molecular cascade. Lactylation with loss of HIF-1 α induces additional senescence and glycolysis, disrupting autophagy, OXT, and EPO. In addition, it promotes premature aging, thymic involution, ER stress, and ANG II upregulation, engendering a vicious circle of hypoxia-cellular senescence-lactate—and further senescence.

**Figure 2 biotech-12-00038-f002:**
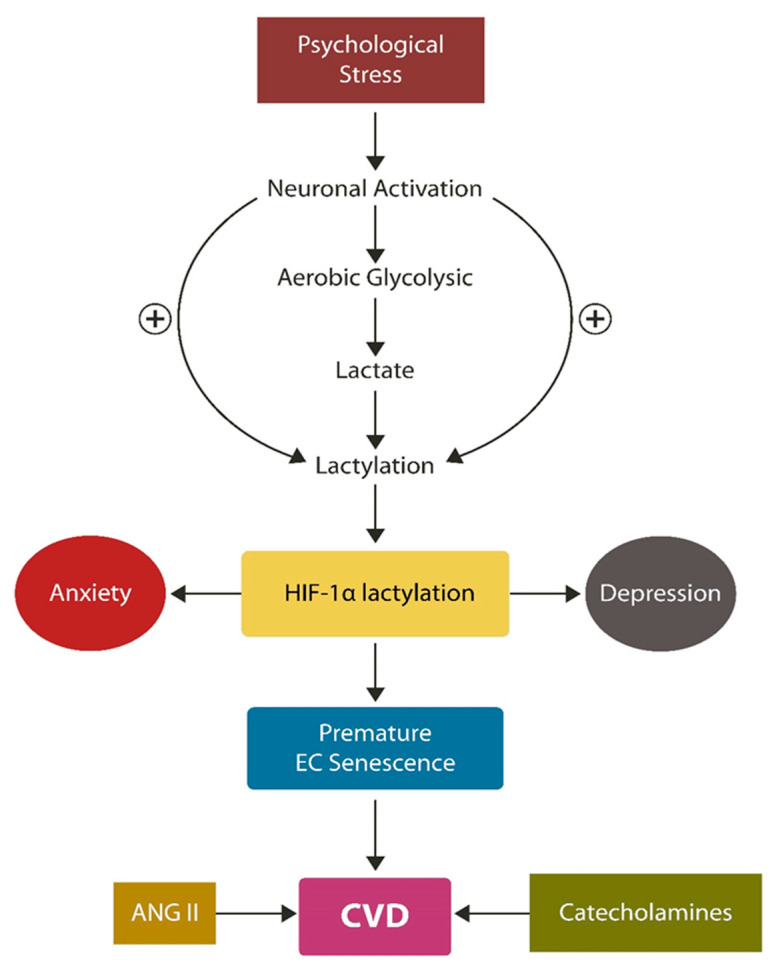
A closer look at the PTSD/CVD connection: overwhelming psychological stressors activate the stress-processing brain areas, promoting glycolysis and excessive lactate. Lactate serves as a lactylation substrate, altering the function of many proteins, including HIF-1 α. Loss of HIF-1 α promotes hypoxia with subsequent premature endothelial senescence, depression, and anxiety. Premature endothelial senescence, stress-mediated release of catecholamines, and ANG II may trigger CVD.

**Figure 3 biotech-12-00038-f003:**
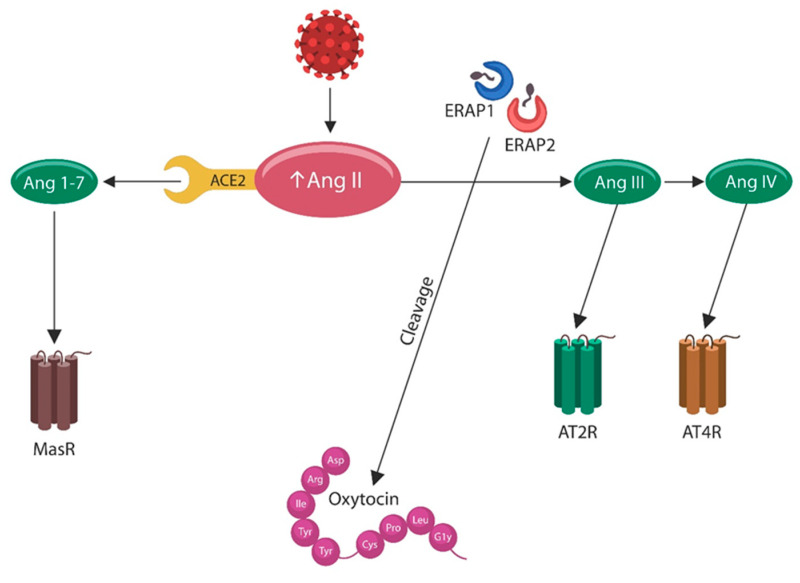
The SARS-CoV-2 virus attachment to ACE-2 disables this enzyme, leading to ANG II accumulation. ERAP1 And ERAP2 are complementary to ACE-2 as they hydrolyze ANG II into the protective ANG III and ANG IV. Loss of ERAP2 (in approximately 25% of the population) slows the processing of ANG II and OXT, increasing the susceptibility to some pathologies, such as hypertension, autoimmunity, PTSD, and CVD.

**Figure 4 biotech-12-00038-f004:**
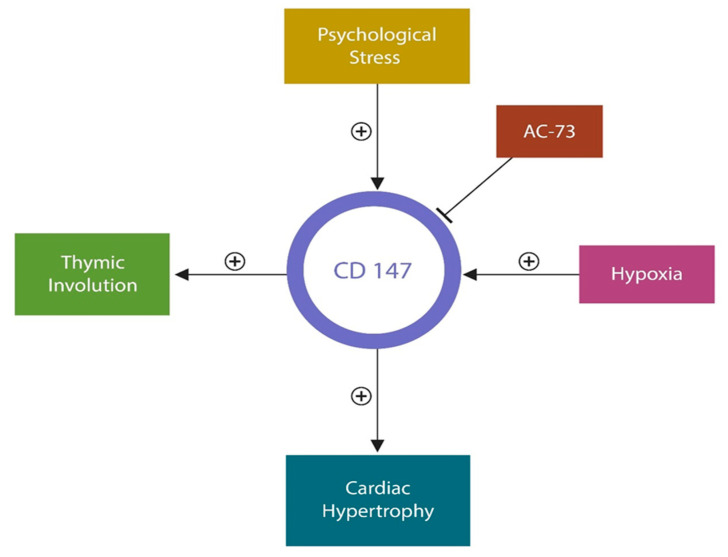
ac-73 is a basigin (cd147) inhibitor that could be beneficial to patients with ptsd and cvd. hypoxia, thymic involution, and autoimmune inflammation upregulate cd47. In addition, psychological stress upregulates CD147, promoting CVD and senesce-related pathology.

**Table 1 biotech-12-00038-t001:** Physiological and pathological Il-7 signaling.

IL-7 Signaling	PTSD/CVD	References (Adapted from)
Upregulate Warburg effect	Upregulate Warburg effect	[153]
Lower SARS-CoV-2 risk	Higher SARS-CoV-2 risk	[154,155,156]
Thymic activation	Thymic autophagy	[65,157]
Upregulated autophagy	Downregulated autophagy	[158,159,160,161]
Lower endothelial/immune senescence	Increased endothelial/immune senescence	[44,162]

**Table 2 biotech-12-00038-t002:** Potential interventions for PTSD and CVD.

Therapeutic Agent	Mechanism	References (Adapted from)
OXT		
Intranasal OXT	Substitution therapy	
*Lactobacillus reuteri*	Synthesizes OXT	[196,197]
HIF-1α		
Intermittent hypoxia	Upregulates HIF-1α	[8,9,10]
Hyperbaric oxygen	Lowers hypoxia	[11,12,13,14,15,16]
A2B blockers		
PSB-603	Decrease obesity and aging	[203,204]
ASIC blockers		
Amyloride	Potassium-sparing diuretic	[214,215]
C5B	Neuroprotective	[216]
Sevanol	ASIC1 and ASIC3 inhibitor	[219,220]
Epigallocatechin gallate	ASIC3 inhibitor	
LDH inhibitors		
5-ALA	Improves mood, fatigue, sleep	[229,230,231]
Quinoline 3-sulfonamides	Reverses glycolysis	[233,234]
Galloflavin	Inhibits lactate production; decreases ATP synthesis	[217]
Autophagy activators		
Antidepressants/ketamine	Increase autophagic markers LC3II/I	[242,243,244,245]
Recombinant ERAP2	Lower ER stress	[246]
Lactylation inhibitors		
Demethylzeylasteral (DML)	Lowers depression, atherosclerosis, pain	[238]
IL-7		
IL-7 recombinant GX-17	Prevents cellular senescence	[247,248]
Thymic activators		
Photobiomodulation	Restores thymic function	[67]

## Data Availability

The study did not report any data.

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
