# Peer review of "Recent Developments in Protein Lactylation in PTSD and CVD: Novel Strategies and Targets"

_biotech, 2023, doi:10.3390/biotech12020038_

Round 1
Reviewer 1 Report
The current manuscript aimed to explore the relationship between PTSD and CVD specifically through hypertension and protein lactylation. While there has been a scientific discussion linking diseases such as PTSD and risk for dementia, there has been a lack of understanding the biological reason for this, so the review is timely and an important topic. It is nicely organized and well-written, but there are a few issues that should be resolved prior to publications.
· The authors focus on lacylation and oxygen-independent energy consumption in brain endothelial cells but switches to a discussion to more specific regions with the brain parenchyma (e.g. amygdala). However, neural and glial cell metabolosis is very different then endothelial cells. Please elaborate on region-specific lacylation mechanisms.
· The authors discuss ERAP2 function in the context of PTSD but does not provide a strong link to CVD. For example, what would happen if ERAP was blocked, would a patient be more likely to develop CVD?
· More connections between LDH inhibitors and CVD/PTSD should be made, if the authors are arguing that PTSD and CVD are biologically related.
· Some minor grammatical edits:
Line 15- Extra space before "However" needs to be deleted
Line 185- Add comma after viPTSD
Line 187- Delete space before "For"
Line 208- Neurotrophin spelled incorrectly
Line 210- Extra space before 's' needs to be deleted
Line 213- Add period after requirements and capitalize H in 'however'
Line 257- Remove comma after severe
Line 346- Add semicolon after studies
Author Response
Thank you very much for your constructive comments. We have responded as follows:
- All minor grammatical edits have been implemented as indicated
-
The authors focus on lacylation and oxygen-independent energy consumption in brain endothelial cells but switches to a discussion to more specific regions with the brain parenchyma (e.g. amygdala). However, neural and glial cell metabolosis is very different then endothelial cells. Please elaborate on region-specific lacylation mechanisms.
Response: Endothelial cells are 80% dependent on aerobic glycolysis and rarely utilize oxidative phosphorylation (Du W). Upon activation, microglia shift their metabolism from oxidative phosphorylation to aerobic glycolysis (Cheng, J). Astrocytes are primarily glycolytic and generate lactate that is shuttled to the neurons for energy needs (Mason S). Neuroimaging has demonstrated that activated areas of the brain upregulate aerobic glycolysis, while aging brains decreases this metabolic modality (Goyal MS).
Elevated LDH is a risk factor for both PTSD and CVD (measured by Framingham risk score). However, in Alzheimer’s disease there is lowered aerobic glycolysis and for this reason, it may be counterproductive to inhibit glycolysis completely (risk of neurodegeneration). Demethylzeylasteral (DML) seems to selectively lower lactylation of histone proteins (epidrug) more than lactate itself (Pan L). The target in PTSD and CVD is post-translational modification, lactylation, an epigenetic mechanism. For this reason, LDH inhibitors may not be the ideal therapy. Lowering cellular senescence, reversing thymic involution, enhancing EPO and IL-7 lower histone lactylation. For example, senolytic drugs, including Dasatinib, quercetin, and rapamycin, oppose cellular senescence a phenotype known to upregulate lactylation (Dookun E)( Wróbel-Biedrawa D). Other approaches, include acid-sensing ion channel-1a (ASIC1a) inhibition and PDK1 inhibitors, agents that can reverse cellular senescence and showed beneficial effects in CVD and PTSD (Barile E).
Since we described these treatments in other articles on PTSD in the context of COVID-19, we have not repeated them here (Sfera A). - The authors discuss ERAP2 function in the context of PTSD but does not provide a strong link to CVD. For example, what would happen if ERAP was blocked, would a patient be more likely to develop CVD? Response: ERAPs are protective against hypertension, an important risk factor for the development of stroke, heart failure, and cardiovascular and renal disease (Zee RYL). Therefore, as ERAPs cleave ANG II, terminating its action, blocking these aminopeptidases would increase the odds of developing hypertension that in return increases the risk of CVD (Mattorre B).
- More connections between LDH inhibitors and CVD/PTSD should be made, if the authors are arguing that PTSD and CVD are biologically related. Response: Please see response above and section within the manuscript "Significance for the field and novel targets", where such links are further strengthened.
Reviewer 2 Report
After a careful review of this manuscript, I fail to understand the motive of the article. It is just another review article in the ocean of several reviews. Authors need to justify why and how this review will be beneficial for the researcher in the field. They need to mention a separate section in this article and address that.
I can not recommend this article for publication in its current form.
Author Response
Thank you for your comments.
PTSD was first described as “shell shock” in WWI. Not much progress has been made in the treatment of this disorder since. I am a psychiatrist and have been working with US veterans since 1996. I hear their complaints and empathize with their suffering. Anything that would help them, would be welcome.
Lactylation is a newly defined posttranslational modification, and, at least to my knowledge, has not been considered in the pathogenesis of PTSD (not according to the Veteran Administration published studies). First and foremost, any potential new treatment for this disorder is more than welcome.
Moreover, we are arguing here that PTSD is a systemic illness, involving premature endothelial aging throughout the body. For this reason, novel approaches should include systemic interventions, like epigenetic treatments. For example, the therapies we have today, primarily antidepressants, are mostly counterproductive as they induce more vivid dreams that usually increase the nightmares. Prazosin makes them tired and rarely takes care of nightmares. For these reasons, we need new approaches and we are pointing here to epigenetics.
As reviewer 1 had related suggestions, we have added the following paragraphs (highlighted in yellow within the manuscript - and copied in full below).
Significance for the field and novel targets
Although identified as a pathology during WWI when PTSD was known as “shell shock”, today, there is a paucity of treatments for this condition (267). The PTSD cardiovascular involvement was hinted at by the term “soldier's heart” used even earlier, during the American Civil War, to describe feelings of anxiety, tension, and being “on edge”.
We have discussed in other articles premature endothelial senescence in PTSD and advocated for different approaches than those utilized today that lack efficacy in most patients (268). For this reason, we focus primarily on ECs as they derive 80% of energy from aerobic glycolysis and rarely utilize oxidative phosphorylation (OXFOS) (269). In contrast, other brain cells, such as resting microglia rely on OXFOS, while activated microglia shift their metabolism to aerobic glycolysis (270). Astrocytes are primarily glycolytic and generate lactate that is shuttled to the neurons for their energy needs (271). Neuroimaging studies have demonstrated that activated areas of the brain upregulate aerobic glycolysis, while aging brains decreases this metabolic modality (272). In PTSD, due to premature cellular aging, there is less activation or even deactivation in ventromedial prefrontal cortex during trauma-induced script-driven imagery (273).
Although treated primarily by psychiatrists, PTSD is a systemic disease which has been associated with numerous medical problems, including chronic pain, hypertension, fatigue, muscle tension, headaches as well as metabolic syndrome and immune dysfunction (274)(275)(276). As systemic disorders call for systemic interventions, epigenetic therapy to lower lactylation of histone proteins seems to be an adequate approach.
The treatment targets, including lowering cellular senescence, reversing thymic involution, enhancing EPO and IL-7, may lower lactylation without totally inhibiting lactate, a metabolic modality utilized by healthy brains. For example, senolytic drugs, including dasatinib, quercetin, and rapamycin, oppose cellular senescence, a phenotype known to upregulate lactylation, especially in ECs (277)(278). Other systemic approaches, include ASIC1a and 3-phosphoinositide-dependent protein kinase 1 (PDK1) inhibitors, agents that can reverse cellular senescence and showed beneficial effects in CVD and PTSD (279).
Round 2
Reviewer 2 Report
Thanks for the justification.
The newly added section in the paper made this review more relevant.